# The relationship between body condition, body composition, and growth in amphibians

**Ross K. Hinderer** [iD][1]*, **Blake R. Hossack**[1,2], **Lisa A. Eby**[1]

**1** Wildlife Biology Program, Franke College of Forestry and Conservation, University of Montana, Missoula, Montana, United States of America, **2** U.S. Geological Survey, Northern Rocky Mountain Science Center, Missoula, Montana, United States of America

These authors contributed equally to this work.
* rkhinderer@gmail.com

## Abstract

Body condition of animals is often assumed to reflect advantages in survival or reproduction, but body condition indices may not reflect body composition, or condition may be unrelated to fitness-associated traits. The relationship between body condition indices and composition has rarely been quantified in amphibians, and body condition has not previously been related to growth in adult amphibians. We used laboratory (quantitative magnetic resonance) and field methods to evaluate the relationship between body composition and the four common body condition indices for wildlife studies (body mass index, Fulton's index, scaled mass index, and residual index) in two frog and one salamander species in Montana, USA. We then assessed the relationship between body condition and summertime somatic growth during a 3-yr mark-recapture study of one of our study species (Columbia spotted frogs, *Rana luteiventris*). Correlation of body condition indices with fat and lean mass differed across species, sexes, and whether components were represented as percentages or were scaled based on size. Scaled mass index, residual index, and Fulton's index were most often well correlated ($r > 0.6$) with scaled body components, but Fulton's index was strongly correlated with body length. Scaled mass and residual indices predicted scaled fat relatively well and were uncorrelated with body length. Heavier condition predicted higher growth rates of Columbia spotted frogs, regardless of the index used. Frogs of heavy body condition (90th percentile residual index) grew 0.04 and 0.05 mm/day greater than frogs of light condition (10th percentile) for average length males and females, respectively. Frogs of short body length (10th percentile) grew 0.11 and 0.19 mm/day more than long (90th percentile) males and females, respectively. By examining the relationship between body condition indices and body composition and revealing a link between condition and future growth, our results provide an empirical basis for choosing the most appropriate condition index, as well as a potential link to fitness-related traits.

## Introduction

The study of individual fitness, or advantages in survival and reproduction of one individual over another, has been a focus of biologists since the formulation of the theory of evolution [1].

**Data availability statement:** Data for this study are available on Figshare in two separate DOIs. https://doi.org/10.6084/m9.figshare.28578131.v1; https://doi.org/10.6084/m9.figshare.28578134.v1

**Funding:** RKH was supported by University of Montana BRIDGES (https://www.umt.edu/bridges/) through funding from the National Science Foundation under Grant No. DGE-1633831, and by a fellowship from the University of Montana Graduate School (https://www.umt.edu/grad/). Funders played no role in study design, data collection or analysis, preparation, or decision to publish this work.

**Competing interests:** The authors have declared that no competing interests exist.

Fitness can be measured in several ways, but the basic components are survival (and longevity) of an individual and reproductive output, as compared to conspecifics [2–5]. Because populations grow through contributions of individuals as they survive and reproduce, differences in fitness can affect population dynamics [6,7]. Direct measurement of survival and reproduction, when possible, is typically labor and time intensive. However, individual traits which reflect survival or reproduction advantages can be an effective proxy for fitness [8].

Body condition, the mass of an animal relative to its size, is often used as an indicator of physiological state [e.g., ability to survive and reproduce; 9] and is readily available from field measurements. Body condition indices have long been used as proxies for fitness in wildlife [10–12], and can also indicate habitat quality [13,14]. More rapid growth or greater size could confer advantages in survival and fecundity, and body condition has been related to growth in fish and reptiles [15–17]. If body condition indicates potential for growth in other vertebrate ectotherms it could be a reliable indicator of fitness [*sensu* 18].

The assumed benefit of heavier condition is based on relationships between condition and body composition (e.g., the amount of distinct body components such as fat or lean mass in an organism), and between composition and differences in fitness-associated traits. Different condition indices can be related to different elements of body composition across and within species [19–21] and between sexes [21,22]. Therefore, the choice of body condition index may be important for detecting a relationship between body condition and composition. Body condition is typically assumed to reflect fat stores which can increase survival and fecundity of individuals [23–26]. However, condition may not describe fat effectively, or fat may not be an important determinant of fitness [27,28]. Importantly, there can be trade-offs between allocating resources to energy storage or somatic growth [29], so heavier condition may occur at the expense of growth. Other measures of body composition, such as lean mass, may also confer fitness benefits through hydration, energy storage, or increased locomotor performance [30–33]. But outside of a few relatively well-studied taxa, there is little understanding of the mechanism underlying the relationship between condition measured from length and mass, the underlying body composition of an animal, or how condition could translate to fitness differences.

Body condition is appealing for providing insight into fitness or population growth of species that are rare or imperiled, such as many amphibians, because it is a non-destructive metric that is measurable among individuals in the field. The relationship between body condition and fitness-related traits, such as growth, survival, and fecundity, remains poorly understood in amphibians. Greater allocation of fat as a percentage of body mass can increase survival in recently metamorphosed amphibians [34], but this relationship remains unexplored in adults. Notably, the relationship between body condition and body composition has rarely been verified [but see 35]. It is also unclear whether condition indices best represent percentage of body fat, or fat mass scaled to body size [35]. Larger size can increase survival of amphibians [36–38], and faster growth can result in more rapidly reaching a "size refuge" from predators [39], or adopting a "faster" life history with earlier time to maturity [40]. If heavier condition indicates potential for growth, then condition could be an indirect indicator of fitness-associated traits; yet, there have been no studies linking body condition and growth in amphibians.

Filling gaps in what body condition represents and whether it is linked with growth can improve our understanding of when body condition is a useful predictor of fitness-related traits. Because there is little information on how body composition is reflected by condition indices, and to our knowledge, no exploration of the relationship between condition and growth in adult amphibians, we sought to measure these relationships. We used lab- and field-based methods to 1) examine the relationship between body condition index and percentage or relative mass of body components (fat and lean mass) and 2) apply body condition indices

to determine whether somatic growth of Columbia spotted frogs (*Rana luteiventris*) was related to condition over 3 years of a capture-mark-recapture study.

## Methods and materials

### Study area and species

We sampled three native species of amphibians (Columbia spotted frog, Sierran treefrog [*Pseudacris sierra*], and long-toed salamander [*Ambystoma macrodactylum*]) from three waterbodies in western Montana, USA. We used animals from all three sites to examine body condition indices, but animals from only one site (Jones Pond) to study growth. Jones Pond is a permanent, small reservoir (1231 m elevation; 46.90°N; -113.44°W) where we captured Columbia spotted frogs and long-toed salamanders; Lost Horse Marsh is a semi-permanent, highly vegetated marsh (1360 m; 46.10N°; -114.27°W) where we captured Columbia spotted frogs, Sierran treefrogs, and long-toed salamanders; and Kramis Pond is a permanent, highly vegetated pond (1316 m; 46.07°N; -114.25°W [S1 Fig in S1 File]) where we captured Sierran treefrogs and long-toed salamanders. We caught all animals by hand and with funnel traps. We found long-toed salamanders, Sierran treefrogs, and very few Columbia spotted frogs during the spring breeding season (May), and we captured most Columbia spotted frogs during summer at Jones Pond (June – August). Though we had no way to assess whether animals had previously bred shortly prior to capture, this is typical in amphibian population studies which occur before, during, or after the breeding season [41]. All three species were used to study body composition; Columbia spotted frogs were also part of a long-term mark-recapture study [42].

### Evaluation of body composition and condition indices

There is little consensus on which index is best suited to describing relative condition or body composition of vertebrates [20,43,44]. Therefore, we evaluated the four most common body condition indices used in wildlife studies (Table 1). We calculated condition indices based on the field-measured snout-vent length ("length", mm) and mass (g).

During 2021–2022, we transported individuals of all three species to the University of Montana's Fort Missoula Research Station for measurements of body composition within 24 hours of capture. Animals were transported in individual, clean plastic containers with a damp paper towel substrate to prevent dehydration, and we observed no visible distress

**Table 1. Body condition indices calculated from mass (*M*) and length (*L*) at measurement occasion *i* of Columbia spotted frogs, long-toed salamanders, and Sierran treefrogs, Montana, USA.**

| Body Condition Index | Formula | Notes |
|---|---|---|
| Scaled Mass Index | $\hat{M}_i = M_i \left[\dfrac{L_0}{L_i}\right]^{b_{sma}}$ | $\hat{M}_i$ is the predicted mass for measurement *i* when scaled to length $L_0$. $b_{sma}$ is the scaling exponent from standardized major axis regression of ln(*M*) on ln(*L*) [19]. |
| Residual Index | $r_i = y_i - \hat{y}_i$ | $r_i$ is the measurement residual from a linear regression of ln(*M*) on ln(*L*). Originally proposed to describe differences in body structure size across taxa [9,45,46]. |
| Fulton's Index | $K_i = \dfrac{M_i}{L_i^{\,3}}$ | $K_i$ is the ratio of mass to length cubed. History of the index is unclear [47]. |
| Body Mass (Quetelet's) Index | $BMI_i = \dfrac{M_i}{L_i^{\,2}}$ | $BMI_i$ is the ratio of mass to length squared. First proposed in 1832 for human health studies, revived in the 1970s [48,49]. |

of any animals during the time (< 24 h) they were held. We scanned all individuals using a body composition analyzer (EchoMRI-500, www.echomri.com, Houston, Texas, USA) based on quantitative magnetic resonance (QMR). Measurements using this machine require no anesthesia or unique sample preparation and do not harm animals. QMR produces measurements of water, lean mass (including muscle and bone), and fat content, in grams per animal. See supplemental information for details of the QMR scanning procedure (S1 Appendix). All protocols complied with Montana Fish, Wildlife, and Parks scientific collector's permit 2021–066-W and University of Montana Institutional Animal Care and Use Committee protocol 014–21LEECS-031621.

To better understand if body condition indices were useful for describing the relative allocation of body components (percent components) or relative mass of body components (scaled components), we calculated both the percentage and scaled mass of fat and lean mass per animal measured with QMR [35,50,51]. The relationship of percent and scaled body fat to length differed markedly between adult and juvenile Columbia spotted frogs (the only species for which we captured juveniles), indicating a difference in fat allocation by size or life stages (S2 Fig in S1 File). To avoid confounding any relationships with a life stage-specific allocation of resources, we removed juvenile Columbia spotted frogs (individuals not showing secondary sex characteristics, which were typically < 45 mm snout-vent length) from all analyses.

We calculated scaled body components (fat and lean mass) per animal using the method of Peig and Green [19]. Scaled mass of a body component was the mass of either fat or lean mass obtained through QMR measurement $i$, multiplied by the ratio of the mean length of that species (across all sites and sexes, adults only) to the actual length of the scanned animal, raised to the power of the scaling coefficient $b_{sma}$ (Table 1):

$$Scaled\ component\ (g)_i = Component\ (g)_i \times \left(\frac{\overline{length}}{length_i}\right)^{b_{sma}}$$

Scaling body components allowed comparison of the effectiveness of indices at predicting body composition at a standardized size per species, reducing the effect of greater raw component mass in larger animals, and the scaling coefficient accounts for the non-linear relationship between body mass and mass of individual components [19,35,51].

We made 117 measurements of adult Columbia spotted frogs, which represented at least 96 individually tagged frogs. Twenty-six Columbia spotted frogs were measured more than once (2–5 times per individual across two years) but never more than once per month. We measured 54 long-toed salamanders and 33 Sierran treefrogs, though none were individually marked. There is a small chance we measured unmarked salamanders and treefrogs more than once, but we suspect it is unlikely as previous research in western Montana showed long-toed salamanders occur in large numbers where they are present [52] and we observed evidence of large numbers of Sierran treefrogs (calling males and egg masses) in the wetlands where we captured animals. We were only able to capture a small number of the likely large populations of these species due to their cryptic nature and terrestrial behavior outside their brief breeding season. The distribution of scaled component measures and percent body components was positively skewed, so we natural log-transformed the values for analyses.

We then examined the correlation (Pearson's $r$) of all four body condition indices with percent and scaled fat and lean mass in each species and sex [35]. We accounted for repeated measurements of some Columbia spotted frogs with a random effect for individual. Because correlation coefficients are not well-defined for mixed effects models, we calculated the square root of Nakagawa's marginal $R^2$ for comparison with the other species [53]. We report results

as weakly correlated ($r < |0.3|$), moderately correlated ($|0.3| < r < |0.7|$), or highly correlated ($r > |0.7|$) in the results.

## Growth of Columbia spotted frogs

To estimate the relationship between body condition and growth of Columbia spotted frogs, we captured frogs at Jones Pond for three or four evenings in each of June, July, and August 2020–2023. We weighed (with digital balance, g), measured length (with ruler, mm), and determined sex (by the presence of nuptial pads in adult males) of captured frogs. We marked all frogs over 36 mm in length by implanting an 8-mm passive integrated transponder (PIT) tag with a unique individual identification number.

Measurements collected during repeated capture events were used to estimate somatic growth. We limited our inference on growth to adult frogs captured multiple times within a summer, excluding long intervening periods of no direct observation and unknown body condition (e.g., individuals captured in June 2021 and not seen again until August 2022). If a frog was captured multiple times within a month, we averaged length and mass measurements during the period. We calculated growth as the change in length between months during the summer of a single year, by individual, in mm/day. This resulted in 234 unique within-summer growth values across the 482 individuals marked. We zero-centered and scaled (SD = 1) all values for body condition index and length to improve model mixing and convergence and allow zero-centering of prior distributions.

We ran one growth model for each body condition index after removing body mass index, which was highly correlated with length across all captured frogs (Table 2). We excluded body mass index to avoid confounding effects of condition with the marginal effect of length on growth. We used linear regressions where growth values were normally distributed with a mean $\mu_{i,t}$ and random error $\sigma_{growth}$ (precision $\tau = 1/\sigma^2$, $\sigma$ uniformly distributed 0–10), for individual $i$ at time $t$ (summer month):

$$Growth_{i,t} \sim N\left(\mu_{i,t}, \sigma_{growth}\right)$$

We modeled the mean growth value $\mu$ as a function of body condition index (BCI), length, sex, and individual,

$$\mu_{i,t} = \alpha_{meanF} + \beta_{BCI} * BCI_{i,t} + \beta_{length} * Length_{i,t} + \beta_{male} * m_i + \varepsilon_i$$

where $m$ was an indicator (0/1) for whether an individual was male. The random effect $\varepsilon$ (mean 0, precision $\tau = 1/\sigma^2$, $\sigma$ prior uniformly distributed 0–5) had $i$ levels to account for

**Table 2. Pearson correlation coefficients _r_ between four body condition indices (scaled mass index [SMI], residual index [Residuals], Fulton's Index [Fulton's], and body mass index [BMI]) and two measures of body fat and lean mass (log percent fat and log percent lean, log scaled fat and log scaled lean) in Columbia spotted frog males (M; _n_ = 48) and females (F; _n_ = 69). The correlation coefficient _r_ is calculated as the square root of the marginal _R_2 value [53] due to the inclusion of a random individual effect. Correlation(SVL) is the correlation between each condition index and snout-vent length (SVL).**

| BCI | Percent Fat | | Percent Lean | | Scaled Fat | | Scaled Lean | | Correlation(SVL) |
|---|---|---|---|---|---|---|---|---|---|
| | M | F | M | F | M | F | M | F | |
| SMI | 0.45 | 0.49 | 0.14 | 0.00 | 0.62 | 0.62 | 0.84 | 0.25 | -0.02 |
| Residuals | 0.46 | 0.52 | 0.17 | 0.05 | 0.63 | 0.65 | 0.86 | 0.21 | 0.00 |
| Fulton's | 0.48 | 0.53 | 0.30 | 0.20 | 0.64 | 0.65 | 0.91 | 0.05 | 0.37 |
| BMI | 0.45 | 0.42 | 0.54 | 0.48 | 0.56 | 0.48 | 0.85 | 0.35 | 0.89 |

repeated growth measurements in some individuals. The intercept term $-_{meanF}$ represented the estimated growth, in mm/day, of a female frog at the mean condition index and length for the entire group of sampled frogs. We used normally distributed prior distributions for all beta coefficients (mean = 0, SD = 31.6).

We implemented the frog growth models using Markov chain Monte Carlo (MCMC) sampling in JAGS [54], accessed through *R* version 4.3.2 [55] and package "R2Jags" [56]. We ran three chains of 10,000 iterations each, with 2,000 iterations discarded as burn-in and no thinning. $\check{R}$ values for all parameters were all < 1.1 and trace plots indicated acceptable mixing of chains.

## Results

### Evaluation of body composition and condition indices

Across all three species, Fulton's Index and body mass index tended to be more correlated with length ($r$ = 0.15–0.89; Tables 2–4) than scaled mass index or residual index. Scaled mass index was weakly negatively correlated with length in long-toed salamanders and Sierran treefrogs ($r$ = -0.20, -0.22; Tables 3 and 4), but not in Columbia spotted frogs ($r$ = -0.02; Table 2). Residual index was weakly correlated or uncorrelated with length in all species ($r$ <|0.11|; Tables 2–4).

The body condition indices most correlated with body fat and lean mass varied based on species, sex, and whether the component was scaled or measured as a percentage. Percent fat was not well reflected by condition indices in male long-toed salamanders ($r$ < 0.10; Table 3) or male Sierran treefrogs ($r$ values all negative; Table 4). Percent fat in both sexes of Columbia spotted frogs was moderately correlated with all condition indices ($r$ = 0.42–0.53; Table 2). The relationship between residual index and body fat percentage varied by month in spotted frogs, which were captured multiple times within a summer (S4 Fig in S1 File). In female

**Table 3. Pearson correlation coefficients *r* between four body condition indices (scaled mass index [SMI], residual index [Residuals], Fulton's Index [Fulton's], and body mass index [BMI]) and two measures of body fat and lean mass (log percent fat and log percent lean, log scaled fat and log scaled lean) in long-toed salamander males (M; *n* = 23) and females (F; *n* = 21). Correlation(SVL) is the correlation between each condition index and snout-vent length (SVL).**

| BCI | Percent Fat | | Percent Lean | | Scaled Fat | | Scaled Lean | | Correlation(SVL) |
|---|---|---|---|---|---|---|---|---|---|
| | M | F | M | F | M | F | M | F | |
| SMI | -0.12 | 0.25 | -0.21 | -0.36 | 0.29 | 0.61 | 0.89 | 0.91 | -0.22 |
| Residuals | -0.11 | 0.38 | -0.18 | -0.46 | 0.30 | 0.71 | 0.90 | 0.85 | 0.01 |
| Fulton's | -0.02 | 0.51 | -0.19 | -0.54 | 0.35 | 0.78 | 0.82 | 0.73 | 0.33 |
| BMI | 0.10 | 0.64 | -0.14 | -0.61 | 0.35 | 0.80 | 0.55 | 0.47 | 0.69 |

**Table 4. Pearson correlation coefficients *r* between four body condition indices (scaled mass index [SMI], residual index [Residuals], Fulton's Index [Fulton's], and body mass index [BMI]) and two measures of body fat and lean mass (log percent fat and log percent lean, log scaled fat and log scaled lean) in Sierran treefrog males (M; *n* = 29) and females (F; *n* = 4, note the small sample size). Correlation(SVL) is the correlation between each condition index and snout-vent length (SVL).**

| BCI | Percent Fat | | Percent Lean | | Scaled Fat | | Scaled Lean | | Correlation(SVL) |
|---|---|---|---|---|---|---|---|---|---|
| | M | F | M | F | M | F | M | F | |
| SMI | -0.01 | 0.31 | -0.35 | -0.45 | 0.39 | 0.76 | 0.98 | 0.99 | -0.20 |
| Residuals | -0.01 | 0.32 | -0.45 | -0.42 | 0.38 | 0.77 | 0.93 | 0.99 | 0.10 |
| Fulton's | -0.01 | 0.34 | -0.45 | -0.38 | 0.37 | 0.78 | 0.91 | 0.99 | 0.15 |
| BMI | -0.02 | 0.39 | -0.50 | -0.26 | 0.19 | 0.80 | 0.43 | 0.99 | 0.59 |

long-toed salamanders, percent fat was weakly to moderately correlated with all condition indices ($r = 0.25–0.64$; Table 3). Percent lean mass was not well reflected by any condition index in male or female long-toed salamanders and Sierran treefrogs ($r = 0.19–0.61$; Tables 3 and 4). In Columbia spotted frogs, percent lean was uncorrelated or weakly correlated with all condition indices except body mass index ($r = 0.54$ [males] and $0.48$ [females]; Table 2).

Scaled fat was moderately correlated with all body condition indices in Columbia spotted frogs ($r = 0.48–0.65$; Table 2) and was moderately to highly correlated with all indices in female long-toed salamanders ($r = 0.61–0.80$; Table 3). Scaled lean mass was highly correlated with all condition indices, except body mass index, in long-toed salamanders, Sierran treefrogs, and male Columbia spotted frogs ($r = 0.73–0.99$; Tables 2–4). Scaled lean mass was weakly to moderately correlated with all condition indices in female Columbia spotted frogs ($r = 0.05–0.35$; Table 2). Residual index, the index least correlated with length, performed similarly to scaled mass index at reflecting scaled fat and scaled lean across all species and sexes (Tables 2–4).

Across all sexes, species, and body components (percent and scaled), scaled mass index, residual index, and Fulton's index most frequently performed well (9 correlations $r > 0.6$), while body mass index performed well less frequently (6 correlations $r > 0.6$; Tables 2–4). Condition indices described scaled body components better than percent body components ($r$ for all scaled components was higher than the equivalent percent component). In all cases but one, scaled fat was better reflected in females than males, whereas scaled lean was better reflected in male than female Columbia spotted frogs and long-toed salamanders. This suggests that sex-linked differences in body composition were reflected in condition indices. In Sierran treefrogs, there was higher correlation between scaled lean mass and body condition indices in females, suggesting species differences in the composition of different sexes; however, this sample size was small ($n = 4$ females; Table 4).

## Growth of Columbia spotted frogs

Based on field measurements, Columbia spotted frogs had growth rates between -0.17 to 0.46 mm/day during June–August (S3 Fig in S1 File). We suspect negative growth values represent measurement error, though there is evidence that amphibians can shrink under drought stress [57]. The three models (one for each condition index except BMI) showed that all body condition indices were positively related to growth (posterior means = 0.02 mm/day more growth with increased condition) whereas length was negatively related to growth (0.09–0.10 mm/day less growth with increased length; Table 5). Fulton's Index was moderately correlated with length of frogs measured in our mark-recapture population ($r = 0.31$), while scaled mass index and residual index were not correlated with length ($r < |0.05|$). Bayesian $p$-values from a posterior predictive check did not indicate a lack of model fit based on the residuals of fitted observations versus data from posterior estimates, for any of the growth models (all $p = 0.47$).

**Table 5. Response-scale posterior distributions for β coefficients and random effect variance (95% credible interval) in three growth models for Columbia spotted frogs (one model for each body condition index), where within-summer growth is a linear combination of scaled body condition index, scaled snout-vent length, sex, and individual (random effect).**

| Growth Model | Intercept | Body condition index | Snout-vent length | Male Sex vs. Female | Individual |
|---|---|---|---|---|---|
| Scaled Mass Index | 0.20 (0.18–0.22) | 0.02 (0.01–0.04) | -0.09 (-0.10 – -0.07) | -0.09 (-0.12 – -0.05) | 0.03 (0.00–0.05) |
| Residual Index | 0.20 (0.18–0.22) | 0.02 (0.01–0.04) | -0.09 (-0.10 – -0.07) | -0.09 (-0.12 – -0.05) | 0.03 (0.00–0.05) |
| Fulton's Index | 0.20 (0.18–0.22) | 0.02 (0.01–0.03) | -0.10 (-0.11 – -0.08) | -0.09 (-0.12 – -0.05) | 0.02 (0.00–0.05) |

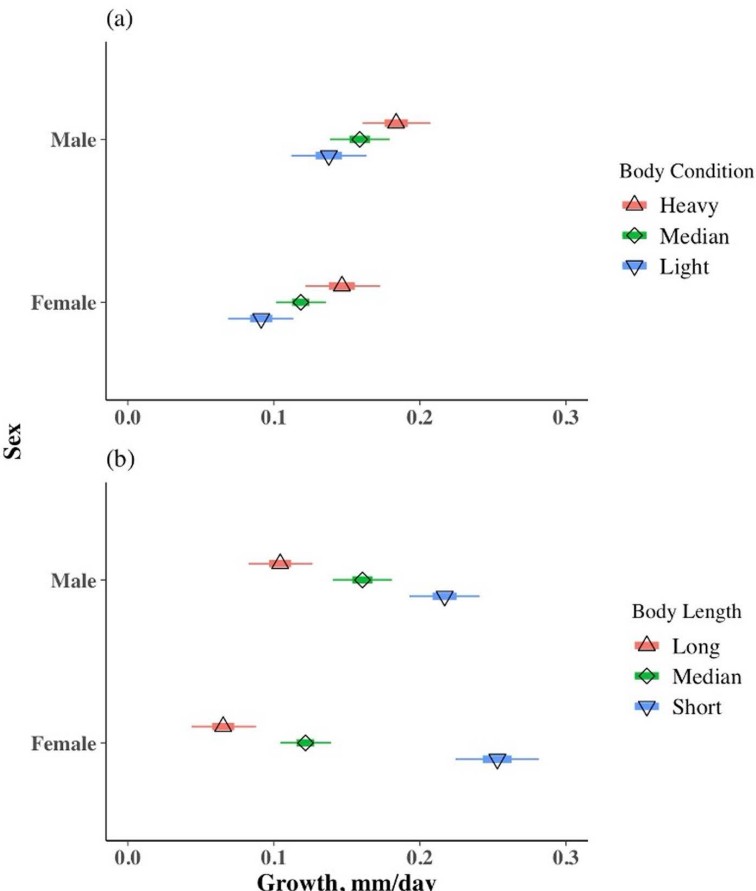

**Fig 1. Posterior distributions of estimated growth values for male ($n$ = 97 growth values) and female ($n$ = 137 growth values) Columbia spotted frogs, (a) at the median snout-vent length observed for the sex and the 90th percentile, median, and 10th percentile residual index observed for the sex and (b) at the mean residual index and the 90th percentile, median, and 10th percentile snout-vent length observed for the sex. Growth was modeled as a linear combination of body condition (residual index), snout-vent length, sex, and individual. Shapes are median posterior values, thick bars are 50% credible intervals, and thin bars are 95% credible intervals. Model results for scaled mass index were similar.**

We estimated how growth varied based on body condition, length, and sex using linear combinations of posterior estimates. Because of the moderate correlation of length with Fulton's Index and the importance of length to growth in all three models, we estimated growth based on residual index, length, sex, and individual. Heavier residual condition index was related to greater growth rate in both sexes (Fig 1; condition distributions in S3 Fig in S1 File). Growth for frogs at heavy body condition (90th percentile) was 0.04 and 0.05 mm/day greater than at light body condition (10th percentile) for an average length male and female, respectively. Small frogs also grew faster than larger frogs. A 10th percentile length frog grew 0.11 and 0.19 mm/day more than long (90th percentile) males and females, respectively, of average condition (Fig 1).

## Discussion

We evaluated the link between body composition and four body condition indices (scaled mass index, residual index, Fulton's Index, and body mass index) in three species of

free-ranging amphibians. By using a combination of lab and field techniques, we described which condition indices derived from field data best indicate fat, a common but rarely verified assumption, and lean mass. By subsequently applying condition indices to capture-recapture study of Columbia spotted frogs, our study revealed increased body condition is related to faster future growth. Our results begin to clarify the complex relationship between body condition and important fitness-related traits.

Overall, condition indices better reflected scaled body components than percentage components across the three species we studied, despite differences in body morphology (e.g., salamanders vs frogs). The percentage of an animal's total body mass dedicated to one component is expected to change as animals grow (e.g., S2a Fig in S1 File), which makes percentage-based measures difficult to compare across sizes, sexes, and developmental stages [19]. For example, percent fat was negatively correlated with all condition indices in male Sierran treefrogs, but the few females in the sample suggested it may be positively correlated in that sex (Table 4). We included percentage components to allow comparison with other, similar studies [e.g., 35], and because a greater allocation of body fat is beneficial to survival in metamorphic amphibians [34]. Typically, however, researchers using condition indices are interested in the relative "heaviness" of an animal for a given size [9], not the allocation of each component to body mass. For the three species of amphibians in our study, all four indices were more correlated with the amount of fat or lean mass for a standardized size (i.e., scaled components).

Several authors have noted that condition and size are commonly correlated in condition indices based on ratios between length and mass, making it difficult to separate the effects of condition and size [9,46,58,59]. Our results validate this concern. Across the three species in our study, Fulton's index was often the most strongly correlated with the greatest number of body components (percent and scaled, all species and sexes). Body mass index and Fulton's Index also tended to increase with SVL ($r = 0.15-0.89$). When examining only the results for scaled fat, usually the desired product of body condition studies, scaled mass index and residual index tended to reflect scaled fat as well as the ratio methods (maximum difference in $r = 24\%$) while also being uncorrelated (residual index) or weakly, negatively correlated (scaled mass index) with length. These results suggest that, of the four commonly used condition indices we assessed, scaled mass index or residual index are the most suitable for amphibians when scaled fat is a quantity of interest.

The relationship between condition indices and components often varies among species, methods of calculation, or sex and size within a species [19,21,51]. We found the relationship between condition and all body composition metrics varied by species and sex, but some important patterns emerged nonetheless. In all but one instance (body mass index of Columbia spotted frogs), condition indices better indicated scaled fat of females than of males. This suggests females tend to add more fat than lean mass when gaining weight, which may reflect the energy reserves required for egg production. On the other hand, males added lean mass, which may help improve locomotor performance or mating success [60]. Accordingly, scaled lean mass was better correlated with all condition indices in males of two species (Columbia spotted frogs and long-toed salamanders). In the third species (Sierran treefrogs), the small sample of females ($n = 4$) might have obscured this pattern. This variation in body composition across species and sexes underscores the importance of evaluating indices of body composition, especially if assuming the indices provide insight into any particular component [35,43].

Although the relationship between body condition indices and body components varied widely, there was strong concordance in the direction of the relationship between condition indices and growth. Heavier body condition was related to faster future growth by Columbia spotted frogs across all three growth models (0.02 mm/day more growth with a 1-SD increase

in condition), indicating animals with heavier condition could attain larger sizes faster than lighter condition conspecifics. To our knowledge, no previous research has linked body condition to growth in adult amphibians. There are many potential fitness benefits of larger adult size, such as larger clutches [61,62] and greater mating success [63,64]. However, faster growth may not directly translate to increased fitness. Because senescence is plastic in amphibians [65], increased growth rate and faster senescence might indicate a shift to a faster life history, which often coincides with smaller maximum size and earlier mortality [66]. Past studies have shown both positive [e.g., 67–69] and neutral [70–72] relationships between body condition and amphibian survival. Heavier condition and faster growth could provide another avenue to increased fitness through shortened generation times or increased fecundity, even if survival is unrelated to condition.

The strong relationship between size and growth limited the utility of the ratio-based indices (body mass index and to a lesser extent, Fulton's Index) to predict future growth of Columbia spotted frogs, because these indices were correlated with length (Table 2). These indices obscured the important, opposing relationships between faster growth at heavier condition and shorter lengths. For example, even though being male was negatively related to growth, at their respective median lengths, male Columbia spotted frogs grew approximately 0.05 mm/day more than females (Fig 1), because they were typically smaller (S3 Fig in S1 File). For this reason, the condition indices uncorrelated with SVL (scaled mass index, residual index) are better suited to describing growth potential in Columbia spotted frogs.

Body condition of Columbia spotted frogs varied within summers, which could have important implications for understanding the relationship between condition and fitness. Typically, researchers have attempted to link annual body condition measurements to survival [67,68,72], the most readily observed aspect of fitness in long-term, mark-recapture studies. We suspect the timing of measurements heavily influences the perceived relationship between body condition and survival, or between body condition and body composition. Though we did not estimate survival, female Columbia spotted frogs tended to have a higher percentage of body fat in August, compared to June or July (S4 Fig in S1 File). Therefore, similar to results from studies of birds and mammals [24,73], a one-time measurement of amphibian body condition may not adequately reflect resources available for survival, or when resources are limited to the point where fat stores become important metabolic fuel. This could explain why most previous research has generally failed to support a positive relationship between body condition and survival in amphibians [70–72,74,75].

Directly measuring survival or reproduction of individuals is often difficult or impossible, making indicators of individual fitness such as body condition important to understanding which individuals are most likely to contribute to population growth. However, the relationship between body condition and body composition or fitness-related traits is often unknown. By evaluating the relationship between body condition indices and body composition and revealing a link between condition and future growth, our results provide an empirical basis for choosing the most appropriate condition index as well as a link to potential fitness. Our results also suggest simple field measures of body condition could help provide insight into at-risk populations of amphibians and other imperiled species by better understanding of an often-used indicator of fitness.

## Supporting information

**S1 File. Supplemental information including S1 Appendix, S1 Figure, S2 Figure, S3 Figure, and S4 Figure. S1 Appendix**. Details of quantitative magnetic resonance protocol for determining body composition in 3 species of amphibians. **S1 Fig.** Locations of capture

sites in Montana, USA for studying body composition of three species of amphibians (all sites; square, circle, and triangle), as well as growth of Columbia spotted frogs (Jones Pond only; square). Made using Natural Earth. **S2 Fig.** Percent body fat (a) and scaled body fat (b) as measured by quantitative magnetic resonance, by snout-vent length in Columbia spotted frogs captured in Montana, USA. The relationship (Pearson's correlation $r$) of percent fat and scaled fat to snout-vent length varied markedly between adults (males [M] and females [F]) and juveniles (unknown sex [U]), so juveniles were removed from analyses of body composition and growth. Shaded areas are 95% confidence intervals for the linear relationship. **S3 Fig.** Distributions of growth in mm/day, snout-vent length (SVL), and four body condition indices (body mass index, scaled mass index, residual index, and Fulton's index) as measured in Columbia spotted frogs in summers of 2020–2023 ($n = 743$ SVL and body condition measurements, $n = 234$ growth measurements). **S4 Fig.** Percent body fat by residual index of female (a) and male (b) Columbia spotted frogs at Jones Pond, Montana. Measurements of body fat were taken via quantitative magnetic resonance across three months in summers 2021 and 2022. Shaded areas are 95% confidence intervals for the linear relationship.
(DOCX)

## Acknowledgments

The authors would like to thank B. Tornabene and L. Fischer for field assistance, and C. Wolf and Z. Cheviron for help with the quantitative magnetic resonance protocol. Comments from Z. Cheviron, W. Lowe, P. Lukacs, C. Pearl, and three anonymous reviewers improved this manuscript. This is US Geological Survey Amphibian Research and Monitoring Initiative (ARMI) product no. 945. Any use of trade, firm, or product names is for descriptive purposes only and does not imply endorsement by the U.S. Government.

## Author contributions

**Conceptualization:** Ross K. Hinderer, Blake R Hossack, Lisa A Eby.

**Data curation:** Ross K. Hinderer.

**Formal analysis:** Ross K. Hinderer.

**Funding acquisition:** Ross K. Hinderer.

**Investigation:** Ross K. Hinderer.

**Methodology:** Ross K. Hinderer, Blake R Hossack, Lisa A Eby.

**Project administration:** Ross K. Hinderer, Blake R Hossack, Lisa A Eby.

**Resources:** Blake R Hossack, Lisa A Eby.

**Supervision:** Blake R Hossack, Lisa A Eby.

**Visualization:** Ross K. Hinderer.

**Writing – original draft:** Ross K. Hinderer.

**Writing – review & editing:** Ross K. Hinderer, Blake R Hossack, Lisa A Eby.

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
