## [Decision Letter · Decision Letter 0]

8 Oct 2024

PONE-D-24-41754The relationship between body condition, body composition, and growth in amphibiansPLOS ONE

Dear Dr. Hinderer,

Thank you for submitting your manuscript to PLOS ONE. After careful consideration, we feel that it has merit but does not fully meet PLOS ONE’s publication criteria as it currently stands. Therefore, we invite you to submit a revised version of the manuscript that addresses the points raised during the review process.

We look forward to receiving your revised manuscript.

Kind regards,

Juan Scheun

Academic Editor

PLOS ONE

Journal Requirements:

Reviewers' comments:

Reviewer's Responses to Questions

**Comments to the Author**

1. Is the manuscript technically sound, and do the data support the conclusions?

Reviewer #1: Partly

Reviewer #2: Yes

2. Has the statistical analysis been performed appropriately and rigorously? 

Reviewer #1: Yes

Reviewer #2: Yes

3. Have the authors made all data underlying the findings in their manuscript fully available?

Reviewer #1: No

Reviewer #2: Yes

4. Is the manuscript presented in an intelligible fashion and written in standard English?

Reviewer #1: Yes

Reviewer #2: Yes

5. Review Comments to the Author

Reviewer #1: The manuscript presents a simple but very useful study linking body condition with body composition. While the results and recommendations are useful, I have several concerns regarding the study in general:

1. the sampling period was large covering pre and post reproduction. During reproduction both females and males lose weight. This will bias the results. Please clarify this aspect

2. the sample size for females in treefrogs is very small (4 individuals) to draw any sound conclusions.

3. the fact that Fulton's index and BMI are correlated with SVL is not unexpected. This is the main reason the two indeces are seldom used (at least to my knowledge).

4. considering that a single species of salamander and only two anuran species were used, the conclusion in L292-3 is totally speculative and not supported by the data.

My suggestion is to limit the analysis to Columbia spotted frog data. This will provide a solid case study. It could afterwards include other species and allow or a search of patterns and trends.

Reviewer #2: This study aims to link 4 values of body condition indices, which are calculated from values easily measurable in the field, to the body composition in 3 species of amphibians, and then test the reflection of these BCIs in the future growth of one of the species. The authors do a good job explaining the importance and goals of their research, describing in detail their methods and results. Following long-term growth rates of individuals in their natural environment is difficult and as such not many studies are available, and the connection with BCI provides us a better choice between various indices.

I have two main comments, which should be easily fixable:

- First, although the study was conducted “during the breeding season through late summer” (line 87), we are given no information related to how data of females full of eggs were treated. Since egg-mass can represent an important part of body mass before egg-laying (even 70-80%), this is likely to introduce an important bias into the values of BCI. An additional related question is: were such animals scanned, and to what type of component would the egg-mass belong? If the answer is simply that all individuals were measured after reproduction ended, this should be clearly stated in the ms.

- There are some unclear/confusing aspects regarding juveniles/ unknown sex throughout the ms. Lines 184 – 185 – the sex does not need to change in the model, you should assign the correct sex to the whole history of the individuals once you know it, before analysis. Lines 122-124 – “we removed juvenile Columbia spotted frogs from all analyses” state here the limits of the juvenile definition (SVL under which value). Are juveniles different from unknown sex? Then line 158, “We limited our inference on growth to adult frogs”, but state that in some cases unknown sex was changed to male or female, which suggests that some juveniles were included.

Minor comments

- Lines 5-7 – spell out numbers <10

- Lines 15- 18 – is this analysis done specifically for residual index? State here if so.

- Rephrase line 29-30 “differences in fitness can scale up to population dynamics”

- Rephrase line 62-63

- The fact that BCI has similar patterns in the 3 locations should be tested, either do some tests before analysis, or include the pond of origin as a factor in the models.

- Lines 81-87 – State here also that the growth study was done only in one of the locations, not only in the Fig. 1 legend.

- Line 108 – include here that the animals were also released at capture site within 24 h.

- Lines 125-130 – is the scaling done separately for each sex? State here.

- Figs s1, s2, s4 – mention in legend what the prediction intervals represent. In s2, where no correlation exists, so remove the prediction interval

- Lines 200-201 – are the differences significantly different?

- Table 2 – provide N for males also

- Define at first mention, and then use BCI for body condition index consistently throughout the ms.

- Line 245-248 – this phrase should go in the discussion section, not in the results.

- Line 250 – replace “grew” with “had growth rate values between” or equivalent, to avoid the grew – 0.17 mm /day phrasing.

- Line 251-252 - this phrase should go in the discussion section, not in the results

- Line 252-253 – delete “After removing body mass index due to high correlation with body length”, it is already in the methods. You can just put “(one for each BCI except BMI)” for clarity

- Rephrase line 284-285

- Line 371-373 – this comes out of the blue.

6. PLOS authors have the option to publish the peer review history of their article (what does this mean? ). If published, this will include your full peer review and any attached files.

**Do you want your identity to be public for this peer review?** For information about this choice, including consent withdrawal, please see our Privacy Policy .

Reviewer #1: No

Reviewer #2: No

---

## [Author Response · Author response to Decision Letter 1]

21 Nov 2024

Our responses to each reviewer point are set off by double-carats (>>) below. Line numbers refer to the “clean” manuscript version without track changes. We made some additional edits based on an internal review, which are also reflected in track changes.

Reviewer #1: The manuscript presents a simple but very useful study linking body condition with body composition. While the results and recommendations are useful, I have several concerns regarding the study in general:

1. the sampling period was large covering pre and post reproduction. During reproduction both females and males lose weight. This will bias the results. Please clarify this aspect

>> Sierran treefrogs and long-toed salamanders were captured during their breeding seasons, while Columbia spotted frogs were captured June-August, well after they breed in our area. We had no way to assess whether salamanders or treefrogs had bred prior to capture. However, amphibian surveys often occur during breeding and nonbreeding seasons, and we aimed to make our results useful for practitioners in the field. The fact that some animals may have bred prior to capture should not bias results in any systematic way, as QMR gives a direct measurement of body composition which would reflect any loss of fat, muscle, or water, and loss of mass is reflected in BCI calculations. We have added language to address this point: “Though we had no way to assess whether animals had previously bred shortly prior to capture, this is typical in amphibian population studies which occur before, during, or after the breeding season” (93–95).

2. the sample size for females in treefrogs is very small (4 individuals) to draw any sound conclusions.

>> We agree that the sample size of female treefrogs is quite small. We specifically emphasized this point on lines 261 and 339–340, explaining the lack of relationship observed between allocation of body mass and sex. Elsewhere, both sexes of treefrogs are pooled to avoid making any conclusions from this small sample size.

3. the fact that Fulton's index and BMI are correlated with SVL is not unexpected. This is the main reason the two indeces are seldom used (at least to my knowledge).

>> This is true, and mentioned in the discussion: “Several authors have noted that condition and size are commonly correlated in condition indices based on ratios between length and mass, making it difficult to separate the effects of condition and size” (318–320). We tested the relationship between body composition and these ratio indices, as they are still commonly found in literature from wildlife studies to human health research.

4. considering that a single species of salamander and only two anuran species were used, the conclusion in L292-3 is totally speculative and not supported by the data.

>> We have changed this statement to refer only to the three species in our analysis: “across the three species we studied, despite differences in body morphology” (line 305–307).

My suggestion is to limit the analysis to Columbia spotted frog data. This will provide a solid case study. It could afterwards include other species and allow or a search of patterns and trends.

>> We appreciate this suggestion. However, given the general lack of published research verifying the link between body condition indices and body composition for amphibians, we feel that including all three species is an important contribution to the field.

Reviewer #2: This study aims to link 4 values of body condition indices, which are calculated from values easily measurable in the field, to the body composition in 3 species of amphibians, and then test the reflection of these BCIs in the future growth of one of the species. The authors do a good job explaining the importance and goals of their research, describing in detail their methods and results. Following long-term growth rates of individuals in their natural environment is difficult and as such not many studies are available, and the connection with BCI provides us a better choice between various indices.

I have two main comments, which should be easily fixable:

- First, although the study was conducted “during the breeding season through late summer” (line 87), we are given no information related to how data of females full of eggs were treated. Since egg-mass can represent an important part of body mass before egg-laying (even 70-80%), this is likely to introduce an important bias into the values of BCI. An additional related question is: were such animals scanned, and to what type of component would the egg-mass belong? If the answer is simply that all individuals were measured after reproduction ended, this should be clearly stated in the ms.

>> Sierran treefrogs and long-toed salamanders were captured during their breeding seasons, while Columbia spotted frogs were captured during breeding and during June-August, well after they breed in our area. We had no way to assess whether animals had bred prior to capture. However, amphibian surveys often occur during breeding and nonbreeding seasons, and we aimed to make our results useful for practitioners in the field. The fact that some animals may have bred prior to capture should not bias results in any systematic way, as QMR gives a direct measurement of body composition which would reflect any loss of fat, muscle, or water, and loss of mass is reflected in BCI calculations. We have added language to address this point: “Though we had no way to assess whether animals had previously bred shortly prior to capture, this is typical in amphibian population studies which occur before, during, or after the breeding season” (93–95).

>> We know of no research which has studied the contribution of eggs to whole-body composition in amphibians.

- There are some unclear/confusing aspects regarding juveniles/ unknown sex throughout the ms. Lines 184 – 185 – the sex does not need to change in the model, you should assign the correct sex to the whole history of the individuals once you know it, before analysis.

>> We did not assign sex “backwards” in encounter histories as small individuals were not only unsexed, but also were a separate age class, which were removed before analysis. Assigning a sex to small individuals would have no effect as they were not used in later analysis.

Lines 122-124 – “we removed juvenile Columbia spotted frogs from all analyses” state here the limits of the juvenile definition (SVL under which value). Are juveniles different from unknown sex? Then line 158, “We limited our inference on growth to adult frogs”, but state that in some cases unknown sex was changed to male or female, which suggests that some juveniles were included.

>> We have added language describing what SVL constitutes a juvenile Columbia spotted frog (131–132).

>> We marked and recaptured individuals as they grew across time, which sometimes included the transition from juvenile to adult. We limited our analysis to adult frogs, which included only portions of some animals’ encounter histories. This is the reason for the sex changing throughout an individual’s life (moving from “unknown sex” to male or female) and excluding juveniles, rather than entire individuals.

Minor comments

- Lines 5-7 – spell out numbers <10

>> Throughout the manuscript, we have changed to spelling out numerals <10, as is the PLOS standard.

- Lines 15- 18 – is this analysis done specifically for residual index? State here if so.

>> We have added that that these results are modeled for residual index (line 16).

- Rephrase line 29-30 “differences in fitness can scale up to population dynamics”

>> We have rephrased this sentence: “differences in fitness can affect population dynamics” (now line 31).

- Rephrase line 62-63

>> We have rephrased this sentence: “It is also unclear whether condition indices best represent percentage of body fat, or fat mass scaled to body size” (now 63–64).

- The fact that BCI has similar patterns in the 3 locations should be tested, either do some tests before analysis, or include the pond of origin as a factor in the models.

>> We did not have fully replicated sampling methodology (e.g., a full species x sites design) that would make a comparison of body condition across sites meaningful. Even if there was a difference in body condition index among the three sites, the relationship interest was the allocation of particular components (fat or lean mass) vs condition; light or heavy condition at a particular site would not necessarily change allocation patterns. An analysis of physiological differences in allocation based on environmental differences would be better ascertained by lab studies.

- Lines 81-87 – State here also that the growth study was done only in one of the locations, not only in the Fig. 1 legend.

>> We have added that language: “We used animals from all three sites to validate body condition indices, but animals from only one site (Jones Pond) to study growth” (83–84).

- Line 108 – include here that the animals were also released at capture site within 24 h.

>> We have added the fact that animals were held < 24h here (116).

- Lines 125-130 – is the scaling done separately for each sex? State here.

>> The scaling was across all sexes, as described by Peig and Green. Added language on line 136.

- Figs s1, s2, s4 – mention in legend what the prediction intervals represent. In s2, where no correlation exists, so remove the prediction interval

>> We have added information to each legend that the shaded intervals are 95% confidence intervals for the linear relationship. Keeping the confidence interval, r-value and p-value in S2, even where the relationship is near zero, is more illustrative for the reader.

- Lines 200-201 – are the differences significantly different?

>> Though we do not use formal significance tests for the difference in body fat by month, S4 includes 95% confidence intervals for the relationship between body fat and residual index, and the figure illustrates the general trend we mention in text.

- Table 2 – provide N for males also

>> We have added the N for males and females. The previous version mistakenly included the full sample size in females.

- Define at first mention, and then use BCI for body condition index consistently throughout the ms.

>> We intentionally chose to avoid this acronym because of its similarity to BMI, which caused confusion among early readers of the manuscript.

- Line 245-248 – this phrase should go in the discussion section, not in the results.

>> We feel that this phrase is better suited to the results section as it only reports results from the analysis. Interpretation of these results is included in the discussion paragraph beginning on line 330.

- Line 250 – replace “grew” with “had growth rate values between” or equivalent, to avoid the grew – 0.17 mm /day phrasing.

>> We have changed the language to match this suggestion (264–265).

- Line 251-252 - this phrase should go in the discussion section, not in the results

>> We felt it was important to include this point in the results, as it had little to do with our particular study and readers could be confused by the results if it were not mentioned immediately (265–267).

- Line 252-253 – delete “After removing body mass index due to high correlation with body length”, it is already in the methods. You can just put “(one for each BCI except BMI)” for clarity

>> We agree and have changed the language to match this suggestion (267).

- Rephrase line 284-285

>> We have rephrased the first sentence of the discussion for clarity (297–299).

- Line 371-373 – this comes out of the blue.

>> We have added language to tie this sentence to statements earlier in the paragraph: “by better understanding of an often-used indicator of fitness” (387-388).

---

## [Decision Letter · Decision Letter 1]

15 Jan 2025

PONE-D-24-41754R1The relationship between body condition, body composition, and growth in amphibiansPLOS ONE

Dear Dr. Hinderer,

Thank you for submitting your manuscript to PLOS ONE. After careful consideration, we feel that it has merit but does not fully meet PLOS ONE’s publication criteria as it currently stands. Therefore, we invite you to submit a revised version of the manuscript that addresses the points raised during the review process.

We look forward to receiving your revised manuscript.

Kind regards,

Juan Scheun

Academic Editor

PLOS ONE

Journal Requirements:

Reviewers' comments:

Reviewer's Responses to Questions

**Comments to the Author**

1. If the authors have adequately addressed your comments raised in a previous round of review and you feel that this manuscript is now acceptable for publication, you may indicate that here to bypass the “Comments to the Author” section, enter your conflict of interest statement in the “Confidential to Editor” section, and submit your "Accept" recommendation.

Reviewer #1: (No Response)

Reviewer #3: (No Response)

2. Is the manuscript technically sound, and do the data support the conclusions?

Reviewer #1: Partly

Reviewer #3: Yes

3. Has the statistical analysis been performed appropriately and rigorously? 

Reviewer #1: I Don't Know

Reviewer #3: Yes

4. Have the authors made all data underlying the findings in their manuscript fully available?

Reviewer #1: Yes

Reviewer #3: Yes

5. Is the manuscript presented in an intelligible fashion and written in standard English?

Reviewer #1: Yes

Reviewer #3: Yes

6. Review Comments to the Author

Reviewer #1: The manuscript is improved and most of the methodology is clear now. Nevertheless, I still have some comments.

1. regarding the questions on capture period for Rana luteiventris, in the section Response to reviewers you mentioned " while Columbia spotted frogs were captured June-August, well after they breed in our area" while in the response to reviewer 2 you state "while Columbia spotted frogs were captured during breeding and during June-August, well after they breed in our area." while in the M&M section, L92-93 we mention few during breeding and most afterwards.

L16 and 18 - I suggest to refer only to faster growth rates and not provide the actual values.

L19-21 should be moved at the beginning of the abstract

L41-43 - Please rephrase.

L51 - quantities? Perhaps measures is better

L83 - Please rephrase, you are not validating the indices.

L90 - visual encounter surveys is a method for inventory. Catching is done by hand.

Figure 1 - I suggest moving it to supplementary online

Table 1 - please check this paper that compared three body condition indices and make recommendations on what to use in amphibians

Băncilă, R. I., Hartel, T., Plăiaşu, R., Smets, J., & Cogălniceanu, D. (2010). Comparing three body condition indices in amphibians: a case study of yellow-bellied toad Bombina variegata. Amphibia-Reptilia, 31(4), 558-562.

L148-150 - you mention that the salamanders and treefrogs are abundant, but nevertheless the sample used is rather small. Why?

L211 - Which body condition? It is a bit unclear

L299-300 - please state clear what BCI is better.

Discussion - I suggest to shorten the discussion and limit it to the usefulness of the study. Readers will be interested to see what BCI is recommended, what do they indicate and what are the limits of their use

L422, L477, L543 .... - italics for the scientific name

L511 - clamitans, italics

Reviewer #3: Overall, I agree with the previous reviewers that this study is valuable and will be a useful reference for future amphibian body condition research. In most cases, the authors have adequately addressed the previous reviewers’ comments. However, in a few cases further work is required, largely to provide clarity for readers. I have also included a small number of additional comments that should be addressed before publication.

Previous comments:

Both previous reviewers highlighted that there is an inadequate sample size for female Sierran treefrog (N = 4). Although this is acknowledged by the authors, conclusions and inference are still drawn from these differences. These inferences and conclusions are not supported based on four individuals and therefore, I suggest you only present overall results for this species in Table 4 and in the text (i.e. do not separate the numbers by sex).

Reviewer 2 asked that BCI was defined at first mention, and then to use BCI for body condition index consistently throughout the ms. The authors state that: “we intentionally chose to avoid this acronym because of its similarity to BMI, which caused confusion among early readers of the manuscript.” However, the authors do use “BCI” extensively throughout the results and discussion section (e.g. L233, L235, L237, L238, L347). Whether acronyms are used or not are up to the authors, but consistency is needed in the ms.

Line 211 and Figure S4: Reviewer 2 also commented on this result. Assessing whether body condition differed across months is not an aim of the paper, nor do the authors test this relationship statistically. Figure S4 does not provide a good visual representation of this result because it compares the residual index against percent fat when the comparison should be BCI against month. When focusing solely on the x axis (residual index) the mean residual index looks to be between -0.1 and 0.1 for each month, with some outliers for females and about 0.05 for males. It is not clear if the mean residual index differs between months. I suggest either removing this result and figure or incorporating this question into the manuscript fully. To incorporate it fully, ideally it would be done across the different BCIs (or at least SMI and residual index) and a formal test would be run with the BCI as the response metric and month as the predictor with ‘individual’ used as a random effect.

Reviewer 2 commented on some unclear and confusing aspects regarding the use of juveniles/unknown sex throughout the ms. Regarding the “unsexed” individuals, the authors explain their reasoning in the response to reviewer document. However, this is still confusing in the ms. If juvenile measurements are excluded from all analyses, then Line 192-194 is not required (“we allowed sex to vary by time period because frogs may have grown enough between capture sessions to change from unknown to either adult male or female”) because the “unknown sex” measurements are excluded and only “male” or “female” are therefore contained in the dataset and analysis. Therefore, I suggest removing this sentence for improved clarity.

Additional comments:

Exclusion of juveniles: more justification is needed for removing juvenile individuals, especially for the body component section. The allometric relationship between length and mass is not expected to be linear which is the part of the basis behind the SMI and one of the arguments against the residual index. In the caption of Figure S2 it states: “Juvenile frogs were included in these correlations, but not in body composition analysis or growth models”. Does this mean that juveniles were included in the body condition index calculations but then excluded from on-going comparisons? Either way, this needs to be made clearer in the methods. Furthermore, the residual index and SMI use the relationship between the population mass and population length to calculate their BCI and if only a subset of the population was then used in the body component calculations this may affect the comparison between these measures. I suggest checking that this does not change the inference of your results, and to consider running a sensitivity analysis which does include juveniles in the body component and growth models.

L179: 1 -> one. Also check captions, often using numeric symbols instead of words.

L236: Here and elsewhere in the results the authors use the descriptors “weakly” “moderately” and “highly” but in some cases the R values overlap (e.g. L 246: Moderate = 0.48-0.65, High = 0.61 – 0.8. The authors should set the definition of these descriptors against values within the methods, I suggest near L157.

L251-253: This statement and a similar one in the abstract does not represent the results well and implies that BMI is the better BCI which is not a key result of this ms. Whether or not a BCI ranks better or worse than other BCI is not relevant in cases when all models perform badly. Perhaps more informative would be how often each BCI performed well (e.g. R> 0.6). This would be 5 times for BMI and 9 times for the other three.

L280-283: It is not clear why residual index and not SMI was used for these figures. This should be justified more explicitly, or the figures presented for both BCIs. I would prefer the later, because it aligns better with the overall theme of the ms. to compare BCIs.

L337-339: The BCIs typically correlated well with scaled lean mass except for the large difference between male and female spotted frogs. It would be valuable to discuss potential reasons for this large difference between sex in this species.

7. PLOS authors have the option to publish the peer review history of their article (what does this mean? ). If published, this will include your full peer review and any attached files.

**Do you want your identity to be public for this peer review?** For information about this choice, including consent withdrawal, please see our Privacy Policy .

Reviewer #1: No

Reviewer #3: No

---

## [Author Response · Author response to Decision Letter 2]

25 Feb 2025

PONE-D-24-41754R1

The relationship between body condition, body composition, and growth in amphibians

PLOS ONE

Our responses to each reviewer point are set off by double-carats (>>) below. Line numbers refer to the “clean” manuscript version without tracked changes.

Reviewer #1: The manuscript is improved and most of the methodology is clear now. Nevertheless, I still have some comments.

1. regarding the questions on capture period for Rana luteiventris, in the section Response to reviewers you mentioned " while Columbia spotted frogs were captured June-August, well after they breed in our area" while in the response to reviewer 2 you state "while Columbia spotted frogs were captured during breeding and during June-August, well after they breed in our area." while in the M&M section, L92-93 we mention few during breeding and most afterwards.

>> Thank you for pointing out the inconsistency in our response to reviewers. In the response to reviewer #1, our response should have read “while Columbia spotted frogs were captured during breeding and during June-August, well after they breed in our area…”. The language in the manuscript on lines 91-93 is correct, the response to reviewer #2 is correct, and our justification in the response to reviewers still applies.

L16 and 18 - I suggest to refer only to faster growth rates and not provide the actual values.

>> We appreciate this suggestion, but providing estimated effect sizes in the abstract gives readers a better understanding of the biological significance of the findings. We also feel it is important to illustrate the difference in effect size between body condition and body length’s relationship to growth rate.

L19-21 should be moved at the beginning of the abstract

>> These lines are now at the beginning of the abstract (L 4-6).

L41-43 - Please rephrase.

>> We have restructured this sentence to make the point more clearly (now L 37–39).

L51 - quantities? Perhaps measures is better

>> Agreed, and changed the word “quantities” to “measures” (L 51).

L83 - Please rephrase, you are not validating the indices.

>> We have changed “validate” to “examine” in this sentence (L 83) and throughout the manuscript have used “examine” or “evaluate” instead (L 98, 203, 302)

L90 - visual encounter surveys is a method for inventory. Catching is done by hand.

>> We have changed “visual encounter surveys” to “hand capture” (L 90).

Figure 1 - I suggest moving it to supplementary online

>> Agreed, and this figure (study area map) has been moved to the supplementary material.

Table 1 - please check this paper that compared three body condition indices and make recommendations on what to use in amphibians

Băncilă, R. I., Hartel, T., Plăiaşu, R., Smets, J., & Cogălniceanu, D. (2010). Comparing three body condition indices in amphibians: a case study of yellow-bellied toad Bombina variegata. Amphibia-Reptilia, 31(4), 558-562.

>> Though this study had different goals than ours, we have added a citation in the discussion when discussing the relationship between condition indices and body size in amphibians (L 325).

L148-150 - you mention that the salamanders and treefrogs are abundant, but nevertheless the sample used is rather small. Why?

>> Long-toed salamanders and Sierran treefrogs are difficult to capture in this system. Both species are terrestrial for much of the year and their breeding season is somewhat unpredictable. Sierran treefrogs are highly cryptic and outside the breeding season, though males are more readily captured during breeding due to their calling behavior. We have added additional language to this point on L 147-148.

L211 - Which body condition? It is a bit unclear

>> We have moved this sentence to later in the results, and specified that we meant the relationship between body fat percentage and residual index varied by month in spotted frogs (L 237-239).

L299-300 - please state clear what BCI is better.

>> Though it would be helpful to mention a “best” BCI, our results show that sex, species, and body composition quantity of interest all change which BCI is best reflective of underlying composition. Rather than giving a simple recommendation in the first paragraph of the discussion, we feel the following paragraphs are important to explain the nuance in these relationships.

Discussion - I suggest to shorten the discussion and limit it to the usefulness of the study. Readers will be interested to see what BCI is recommended, what do they indicate and what are the limits of their use

>> As mentioned above, unfortunately there is not a clear recommendation for “best” BCI. Many studies in the past have been comparisons between indices with no discussion of the underlying quantities they reflect, or only focusing on a single species.

We feel that our study, and especially the discussion portion, is an important contribution to the literature surrounding body condition indices and their utility. We do make specific recommendations for BCIs that better reflect scaled fat quantities (L332-334).

L422, L477, L543 .... - italics for the scientific name

>> We have added italic Latin names to the works cited.

L511 - clamitans, italics

>> We have added italic Latin names to the works cited.

Reviewer #3: Overall, I agree with the previous reviewers that this study is valuable and will be a useful reference for future amphibian body condition research. In most cases, the authors have adequately addressed the previous reviewers’ comments. However, in a few cases further work is required, largely to provide clarity for readers. I have also included a small number of additional comments that should be addressed before publication.

Previous comments:

Both previous reviewers highlighted that there is an inadequate sample size for female Sierran treefrog (N = 4). Although this is acknowledged by the authors, conclusions and inference are still drawn from these differences. These inferences and conclusions are not supported based on four individuals and therefore, I suggest you only present overall results for this species in Table 4 and in the text (i.e. do not separate the numbers by sex).

>> We agree that the small sample size of female Sierrran treefrogs prohibits strong inference to that sex. However, evidence from all three species indicates that sexes may exhibit different patterns in the correlation of body components and body condition indices, which could make pooling sexes problematic. We have removed language from the conclusions where we report female treefrog results separately, and only include suggestions of possible sex-linked patterns, where appropriate, while also mentioning the small sample size (L 262–264, L 315–317). We have kept Table 4 separated by sex for consistency across all species, but added a note in the legend to further point out the small female sample size. We feel that, with sample size caveats, it is important to report these findings to potentially spur further research into the sex-linked nature of these relationships.

Reviewer 2 asked that BCI was defined at first mention, and then to use BCI for body condition index consistently throughout the ms. The authors state that: “we intentionally chose to avoid this acronym because of its similarity to BMI, which caused confusion among early readers of the manuscript.” However, the authors do use “BCI” extensively throughout the results and discussion section (e.g. L233, L235, L237, L238, L347). Whether acronyms are used or not are up to the authors, but consistency is needed in the ms.

>> Thank you for pointing out this inconsistency. We have updated our terminology throughout the manuscript, and only use “BCI” for clarity in tables and equations.

Line 211 and Figure S4: Reviewer 2 also commented on this result. Assessing whether body condition differed across months is not an aim of the paper, nor do the authors test this relationship statistically. Figure S4 does not provide a good visual representation of this result because it compares the residual index against percent fat when the comparison should be BCI against month. When focusing solely on the x axis (residual index) the mean residual index looks to be between -0.1 and 0.1 for each month, with some outliers for females and about 0.05 for males. It is not clear if the mean residual index differs between months. I suggest either removing this result and figure or incorporating this question into the manuscript fully. To incorporate it fully, ideally it would be done across the different BCIs (or at least SMI and residual index) and a formal test would be run with the BCI as the response metric and month as the predictor with ‘individual’ used as a random effect.

>> We agree that assessing differences in body composition across months was not an important aim of this paper. We only mention the difference in the relationship (slope in S4 figure) between percent fat and residual index as a general result (L 237–239; the r- and p-values in S4 figure illustrate this difference).

>> We have modified the sentence on L 377–379 to state that percentage body fat of female frogs was generally highest in August, though we did not statistically test this result: “female Columbia spotted frogs tended to have a higher percentage of body fat in August, compared to June or July (S4 Figure)”. We also provide 95% confidence intervals for the relationship in the S4 figure to better visually display differences.

Reviewer 2 commented on some unclear and confusing aspects regarding the use of juveniles/unknown sex throughout the ms. Regarding the “unsexed” individuals, the authors explain their reasoning in the response to reviewer document. However, this is still confusing in the ms. If juvenile measurements are excluded from all analyses, then Line 192-194 is not required (“we allowed sex to vary by time period because frogs may have grown enough between capture sessions to change from unknown to either adult male or female”) because the “unknown sex” measurements are excluded and only “male” or “female” are therefore contained in the dataset and analysis. Therefore, I suggest removing this sentence for improved clarity.

>> We have removed this sentence for clarity.

Additional comments:

Exclusion of juveniles: more justification is needed for removing juvenile individuals, especially for the body component section. The allometric relationship between length and mass is not expected to be linear which is the part of the basis behind the SMI and one of the arguments against the residual index. In the caption of Figure S2 it states: “Juvenile frogs were included in these correlations, but not in body composition analysis or growth models”. Does this mean that juveniles were included in the body condition index calculations but then excluded from on-going comparisons? Either way, this needs to be made clearer in the methods. Furthermore, the residual index and SMI use the relationship between the population mass and population length to calculate their BCI and if only a subset of the population was then used in the body component calculations this may affect the comparison between these measures. I suggest checking that this does not change the inference of your results, and to consider running a sensitivity analysis which does include juveniles in the body component and growth models.

>> We agree that allometry of the length-mass relationship across body sizes is not assumed for the SMI. However, we noted a difference in allocation of body fat as a percentage of mass between juvenile and adult Columbia spotted frogs (L 126-128; S2 Figure). This finding implies a life history difference in growth patterns that could potentially obfuscate important information about the relationship of body condition index to body components. Because we were not able to compare adults and juveniles across all three species, we removed juvenile Columbia spotted frogs from all analyses for consistency.

>> We did not include juvenile Columbia spotted frogs in the calculation of the residual index or scaled mass index. We added language to make this clearer on L 132.

>> We have removed the former S2 Figure from the supplemental material. We agree that it was confusing to portray the correlations of BCIs and SVL across all animals (including juveniles) when juveniles were not included in the analyses.

L179: 1 -> one. Also check captions, often using numeric symbols instead of words.

>> We have changed to spelling out numerals in this line (now L 176) and in figure captions.

L236: Here and elsewhere in the results the authors use the descriptors “weakly” “moderately” and “highly” but in some cases the R values overlap (e.g. L 246: Moderate = 0.48-0.65, High = 0.61 – 0.8. The authors should set the definition of these descriptors against values within the methods, I suggest near L157.

>> We have added a sentence defining our description of correlations (L 155–157) and now follow this convention in the results.

L251-253: This statement and a similar one in the abstract does not represent the results well and implies that BMI is the better BCI which is not a key result of this ms. Whether or not a BCI ranks better or worse than other BCI is not relevant in cases when all models perform badly. Perhaps more informative would be how often each BCI performed well (e.g. R> 0.6). This would be 5 times for BMI and 9 times for the other three.

>> We agree and have incorporated this suggestion into the abstract (L 13-16) and results (L 254-256).

L280-283: It is not clear why residual index and not SMI was used for these figures. This should be justified more explicitly, or the figures presented for both BCIs. I would prefer the later, because it aligns better with the overall theme of the ms. to compare BCIs.

>> We appreciate this suggestion. Results from the growth models of SMI and residual index were extremely similar (Table 5), and the plots were indistinguishable. We include a note in the legend for Figure 1 mentioning the similar results of the two models. We used residual index rather than SMI for the plots because of its broad use in the field, though we acknowledge that SMI would have been appropriate as well.

L337-339: The BCIs typically correlated well with scaled lean mass except for the large difference between male and female spotted frogs. It would be valuable to discuss potential reasons for this large difference between sex in this species.

>> We mention the potential for sex-linked differences in lean mass correlation with BCIs (L 337–342). We note that scaled lean mass was better reflected by BCIs in males of the two species for which we had sufficient sample size of both sexes. As for the marked difference between lean mass correlation in male and female Columbia spotted frogs, we do not think it appropriate in this study to further speculate on the reasons for that difference, besides the reasoning and reference given in lines 337–342.

---

## [Editor Report · Decision Letter 2]

27 Feb 2025

The relationship between body condition, body composition, and growth in amphibians

PONE-D-24-41754R2

Dear Dr. Hinderer,

We’re pleased to inform you that your manuscript has been judged scientifically suitable for publication and will be formally accepted for publication once it meets all outstanding technical requirements. We thank the authors for their effort in answering, or altering, their text.

Kind regards,

Juan Scheun

Academic Editor

PLOS ONE
---

## [Editor Report · Acceptance letter]

PONE-D-24-41754R2

PLOS ONE

Dear Dr. Hinderer,

I'm pleased to inform you that your manuscript has been deemed suitable for publication in PLOS ONE. Congratulations! Your manuscript is now being handed over to our production team.

Kind regards,

on behalf of

Dr. Juan Scheun

Academic Editor

PLOS ONE